# A 3.0 Gsymbol/s/lane MIPI C-PHY Receiver with Adaptive Level-Dependent Equalizer for Mobile CMOS Image Sensor

**DOI:** 10.3390/s21155197

**Published:** 2021-07-31

**Authors:** Seokwon Choi, Changmin Song, Young-Chan Jang

**Affiliations:** 1OLED Design 1 Team, Silicon Mitus, Seongnam 13494, Korea; seokwon.choi@siliconmitus.com; 2Department of Electronic Engineering, Kumoh National Institute of Technology, Gumi 39177, Korea; lasd4529@kumoh.ac.kr

**Keywords:** adaptive level-dependent equalizer, clock recovery, mobile industry processor interface, C-PHY, three-level signal, CMOS image sensor

## Abstract

A 3.0 Gsymbol/s/lane receiver is proposed herein to acquire near-grounded high-speed signals for the mobile industry processor interface (MIPI) C-PHY version 1.1 specification used for CMOS image sensor interfaces. Adaptive level-dependent equalization is also proposed to improve the signal integrity of the high-speed receivers receiving three-level signals. The proposed adaptive level-dependent equalizer (ALDE) is optimized by adjusting the duty cycle ratio of the clock recovered from the received data to 50%. A pre-determined data pattern transmitted from a MIPI C-PHY transmitter is established to perform the adaptive level-dependent equalization. The proposed MIPI C-PHY receiver with three data lanes is implemented using a 65 nm CMOS process with a 1.2 V supply voltage. The power consumption and area of each lane are 4.9 mW/Gsymbol/s/lane and 0.097 mm^2^, respectively. The proposed ALDE improves the peak-to-peak time jitter of 12 ps and 34 ps, respectively, for the received data and the recovered clock at a symbol rate of 3 Gsymbol/s/lane. Additionally, the duty cycle ratio of the recovered clock is improved from 42.8% to 48.3%.

## 1. Introduction

In recent years, high performance in frame rate, color depth and image resolution is constantly required for mobile CMOS image sensors (CISs) [1,2,3,4]. In accordance with this requirement, the mobile industry processor interface (MIPI) C-PHY protocol, along with the MIPI D-PHY protocol, has been studied and reported to increase the interface bandwidth [5,6,7,8,9]. The MIPI C-PHY uses a three-phase symbol with an embedded clock and transmits a three-level signal through three wires per lane. Because of this, unlike the MIPI D-PHY, it does not transmit an additional clock even at high symbol rates, thereby reducing power consumption. The MIPI C-PHY version 1.1 specification supports a symbol rate of 3.0 Gsymbol/s/lane for each lane. It has a total data rate of 20.52 Gb/s when three lanes are used. This value is approximately equal to the total bandwidth when each of the four data lanes of the MIPI D-PHY has a data rate of 5.0 Gb/s/lane. The MIPI C-PHY is mainly used in the short channel between the CIS and the application processor. However, it is necessary that the MIPI C-PHY also support heavy channels for applications such as field-programmable gate array (FPGA)-based frame grabbers used to evaluate CISs, as shown in Figure 1. As the symbol rate or channel loss increases, the signal integrity of the MIPI C-PHY using a three-level signal can be further degraded due to level-dependent inter-symbol interference (ISI) [10].

Continuous linear equalizers (CTLEs) have been widely used to improve ISI in high-speed interfaces because they are implemented using simple circuits [11,12]. In addition, high-speed receivers for the MIPI D-PHY and C-PHY adopted a CTLE to reduce power consumption while improving signal integrity [7,8]. Meanwhile, the high-speed receiver for the MIPI C-PHY reported in a prior study [13] used a level-dependent elastic buffer (LDEB) in addition to a CTLE to reduce the level-dependent ISI. Three elastic buffers and a delay controller were used to implement the LDEB for the high-speed receiver of the MIPI C-PHY, as shown in Figure 2. The LDEB performed time-domain equalization by controlling the delay times of three elastic buffers, respectively.

Figure 3 shows time diagrams to illustrate the operation of the high-speed receiver using the LDEB for the MIPI C-PHY. The signals DA, DB, and DC refer to the three-level signals transmitted from the MIPI C-PHY transmitter. Meanwhile, the signals AB, BC, CA, and CR represent the data and clock signals received and recovered in the MIPI C-PHY receiver. The signals AB, BC, and CA are generated by comparing the voltage differences between DA and DB, DB and DC, and DC and DA. Furthermore, the clock signal CR is recovered by the transitions of the three received data AB, BC, and CA. In particular, when two or three transitions among the three received data occur, the clock signal CR is recovered by the last transition of the three received data, as shown in Figure 3a. Since common mode voltages of the input signals DA, DB, and DC for the high-speed receiver are different from each other, the cross point of the two input signals may be different depending on the voltage level of the input signal; this can generate time jitters of the received data and recovered clock, as shown in Figure 3a. When the cross points are generated at CP_1st_ and CP_3rd_ due to the transitions of three signals with different voltage levels, the time jitters of the received data and recovered clock are generated as the values of T_DDJ_ and T_DDJ_/2, respectively. In this case, time-domain equalization using the LDEB improves the signal integrity from the level-dependent ISI. Figure 3b shows this by applying the LDEB to the signals BC and CA, upon which the time jitter of the received data is reduced to T_DDJ_/2 and the recovered clock, CR, does not ideally contain a time jitter due to the level-dependent ISI. However, as shown in Figure 3c, when excessive delay time is applied to the LDEB, the time jitters of the received data and recovered clock rather increase. Therefore, optimization of the delay time of the LDEB to T_DDJ_/2 is required in the case of Figure 3. Furthermore, the optimal delay time of the LDEB changes according to the loss of the channel. The designed delay time of the LDEB may also not be optimized due to variations in the process, voltage and temperature.

In this work, an adaptive level-dependent equalizer (ALDE), which supports the MIPI C-PHY V1.1 specification, is proposed for application of high-performance mobile camera modules. Furthermore, a pre-determined data pattern, which is transmitted from the MIPI C-PHY transmitter, is proposed to perform adaptive level-dependent equalization. Section 2 presents the proposed adaptive level-dependent equalizer for the MIPI C-PHY receiver; Section 3 presents the measurement results of the implemented MIPI C-PHY receiver, including the proposed adaptive level-dependent equalizer; finally, Section 4 provides the conclusion of this paper.

## 2. Pre-Determined Data for Adaptive Level-Dependent Equalization

During performing the proposed adaptive level-dependent equalization, it is necessary for the pre-determined data to be transmitted from the MIPI C-PHY transmitter to recover the clock signal while including the data-dependent time jitter in the MIPI C-PHY receiver. Meanwhile, the proposed adaptive level-dependent equalization described in Section 3 uses the duty cycle ratio of the recovered clock. The data patterns of the signals DA, DB, and DC in Figure 3 can be used as pre-determined data for the proposed adaptive level-dependent equalization. However, the high region of the recovered clock is adjusted from 1 unit interval (UI) − T_DDJ_/2 to 1 UI + T_DDJ_/2, while the low region of the recovered clock is maintained at 1 UI, as shown in Figure 3. That is, the period of the recovered clock is changed while the adaptive level-dependent equalization is being performed. In this case, even if the recovered clock includes information of data-dependent time jitter, it is difficult for the duty cycle ratio of the recovered clock to be exactly detected because the period of the recovered clock is not fixed at 2 UIs.

The pre-determined data shown in Figure 4 ensure that the recovered clock maintains a constant period of 2 UIs, even if the delay time of the LDEB for the proposed ALDE is changed [14]. According to the change of the delay time for the LDEB, the high and low regions of the recovered clock are adjusted from 1 UI − T_DDJ_/2 to 1 UI + T_DDJ_/2 and from 1 UI + T_DDJ_/2 to 1 UI − T_DDJ_/2, respectively. To perform the optimal operation of the proposed ALDE, the pre-determined data shown in Figure 4 are repeatedly transmitted from the MIPI C-PHY transmitter.

Figure 5 shows the block and time diagrams of the pre-determined data generator used in the MIPI C-PHY transmitter chip for the proposed adaptive level-dependent equalization. The output of the pre-determined data generator, DC[20:0], is supplied to the MIPI C-PHY transmitter instead of the normal data, DP[20:0], to generate the pre-determined data shown in Figure 4 during performing the proposed adaptive level-dependent equalization, as shown in Figure 5a. The signal DC[20:0] is converted to the signals DA, DB, and DC, which are the pre-determined data transmitted through the MIPI C-PHY transmitter for the proposed adaptive level-dependent equalization. The pre-determined data generator shown in Figure 5b is designed by using toggle flip-flops. It begins to generate the signal DC[20:0] after the signal EQCAL_EN, which activates the proposed adaptive level-dependent equalization, becomes high. Figure 5c shows the time diagram of the signal DC[20:0] generated in the pre-determined data generator. The signals DC[18,16,12,10,6,4,0] and DC[19,15,13,9,7,3,1] are the opposite data and have a toggle pattern synchronized with the clock signal CLK_DIV7_, where the signal CLK_DIV7_ is a frame clock for supplying 21-bit parallel data to the MIPI C-PHY Transmitter. The DC[20,17,14,11,8,5,2] are signals with a low level. The block MIPI C-PHY Transmitter is implemented by using circuits reported in previous literature [13].

Figure 6 shows the simulation results of the pre-determined data generator and the MIPI C-PHY transmitter. The frequency of CLK_DIV7_ was set to 428.571 MHz for a symbol rate of 3 Gsymbol/s/lane. Figure 6a shows the three signals DC[2], DC[1], and DC[0] among the 21 DC signals and the three output signals DA, DB, and DC of the MIPI C-PHY transmitter. The signals DC[0] and DC[1], which have opposite data, are toggle data with a data rate of 428.57 Mbps. The pre-determined data had three cross points at different times according to the voltage levels of the three signals, as shown in Figure 6b.

## 3. MIPI C-PHY Receiver Chip

Figure 7 shows a simplified block diagram of the proposed MIPI C-PHY receiver chip. The proposed MIPI C-PHY receiver chip consists of three data lanes supporting high-speed mode and low-power mode operations. Each data lane performs high-speed receive operation and has a bidirectional, low-speed, low-voltage CMOS interface.

To perform the high-speed receive operation of the MIPI C-PHY, each lane consists of a C-PHY RX w/ ALDE, a Clock Recovery, a Two-symbol Decoder, a 3-to-21 Deserializer, a Frame Clock Generator, and a De-mapper. Its basic operation is almost identical to that of the C-PHY high-speed receiver reported in previous literature [13]. From the received high-speed serial data with three voltage levels, each lane of the MIPI C-PHY receiver chip recovers a clock signal (CR) and generates 16-bit parallel data (PO_LANE#_[15:0]) and a frame clock (CLKF _LANE#_). The proposed C-PHY high-speed receiver, C-PHY RX w/ ALDE, performs adaptive level-dependent equalization according to the channel loss caused by the three voltage levels. Furthermore, the De-mapper, which is used to fully implement the MIPI C-PHY V1.1 specification, outputs the 16-bit parallel data from the 21-bit output data of the 3-to-21 Deserializer. It operates in synchronization with the frame clock CLKF _LANE#_, together with the 3-to-21 Deserializer.

### 3.1. MIPI C-PHY Receiver

The proposed MIPI C-PHY receiver consists of three high-speed receivers with a CTLE, an ALDE, and a clock recovery circuit (Clock Recovery), as shown in Figure 8. The proposed ALDE includes a data-dependent time jitter corrector (DDTJC) that determines the delay time of elastic buffers in the conventional LDEB, consisting of three elastic buffers and a delay controller [14]. According to the operation of the LDEB, the Delay Controller determines whether or not to add a certain delay time in the elastic buffer for the next received signals, based on the values of the three currently received and buffered signals AB_OUT_, BC_OUT_, and CA_OUT_. The operation of the LDEB is optimized as the delay time of the elastic buffer is adjusted by the digital control code DEL_CONT[2:0]. According to the concept of the LDEB shown in Figure 4, the data-dependent time jitters of the received data and recovered clock are minimized when the duty cycle ratio of the recovered clock is 50%. In the proposed ALDE, the digital control code DEL_CONT[2:0] is determined by operation of the DDJTC, which controls the recovered clock CR in the clock recovery circuit in order to have a duty cycle ratio of 50%.

### 3.2. Adaptive Level-Dependent Equalizer

The proposed ALDE consists of three elastic buffers, a delay controller, and a DDTJC, as shown in Figure 8. The elastic buffer, delay controller, and clock recovery circuit are designed using circuits reported in previous literature [13].

Figure 9 shows block and time diagrams of the DDTJC. The DDTJC consists of a current integrator, a comparator, a counter, and a control signal generator. It receives the output clock of the clock recovery circuit, CR, as an input signal. First of all, the current integrator generates an analog voltage according to the duty cycle ratio of the clock CR while the control signal CLKI is high. Then, the voltage comparator determines from the result of the current integrator at the rising edge of the control signal CLKC whether the duty cycle ratio of the clock CR is above or below 50%. The result of the voltage comparator is supplied to the counter, and the counter generates the control signal DEL_CONT[2:0] of the ALDE. The current integrator is reset to measure the duty cycle ratio of the updated clock CR when the control signals CLKC and CLKR are both high. The control signal generator generates the synchronization signals CLKI, CLKC, and CLKR for the DDTJC using the clock CR as an input signal.

Figure 10 shows a circuit diagram and operation of the current integrator [15,16]. The current integration is performed by charging the charges to two MOSCAPs of the output nodes V_OP_ and V_OM_ according to the voltage level of the clock CR when the signal CLKI is high. While the clock CR is high, the output node V_OP_ is charged with a slope of four times that of the output node V_OM_. For this, the W/L of M_B1_ is designed to be four times larger than that of M_B2_, as shown in Figure 10a. The integrated output voltages are held when the signal CLKI becomes low. The reset of the current integrator is performed using the signals CLKC and CLKR to minimize switching noise generation. Figure 10b shows that the differential output (V_OP_ ‒ V_OM_) of the current integrator outputs a positive value when the duty cycle ratio of the clock CR is greater than 50%. Conversely, for the clock CR with a duty cycle ratio of less than 50%, the differential output of the current integrator generator is negative, as shown in Figure 10c.

Figure 11 shows the simulation results for the operation of the ALDE when the pre-determined data shown in Figure 4 are supplied to the MIPI C-PHY receiver for the proposed adaptive level-dependent equalization. This simulation was performed at a symbol rate of 3 Gsymbol/s/lane while using transmitter and receiver circuits that support the MIPI C-PHY V1.1 specification. In addition, the pre-determined data generated by the MIPI C-PHY transmitter was transmitted to the MIPI C-PHY receiver through an 80 cm-long FR-4 channel. As the proposed adaptive level-dependent equalization was performed, the output code of the DDTJC, DEL_CONT[2:0], was increased to converge the differential output voltage of the current integrator to zero, as shown in Figure 11a. Actually, Figs. 11b and c show that the differential output voltage of the current integrator decreased from +37.1 mV in PART A to approximately 0 mV in PART B. Furthermore, the duty cycle ratio of the recovered clock CR improved from 53.35% to 49.25%.

The proposed ALDE was evaluated through simulation for arbitrary input data that met the MIPI C-PHY V1.1 specification. This improved the peak-to-peak time jitters of the received data AB_OUT_ and recovered clock CR from 77.7 ps and 50.5 ps to 60.4 ps and 34.3 ps, respectively, as shown in Figure 12. This simulation was also performed under the same conditions as the simulation in Figure 11.

## 4. Implementation and Measurement Results of Proposed MIPI C-PHY Receiver

The proposed receiver chip supporting the MIPI C-PHY version 1.1 specification was implemented using a 65 nm CMOS process with a 1.2 V supply. The total MIPI C-PHY receiver chip area, including three data lanes and parallel output drivers, was 3 mm × 3 mm and each data lane occupied an area of 0.097 mm^2^, as shown in Figure 13. The power consumption of three data lanes was 14.7 mW when the MIPI C-PHY receiver was operated at a symbol rate of 3.0 Gsymbol/s/lane in an 80 cm-long FR-4 channel. The implemented MIPI C-PHY receiver had an energy-per-bit of 2.15 pJ/bit. The MIPI C-PHY transmitter shown in Figure 5 and phase-locked loop (PLL) were implemented together in the MIPI C-PHY receiver chip to generate MIPI C-PHY data as a test block for evaluation of the proposed MIPI C-PHY receiver chip. For this, the MIPI C-PHY transmitter included the pre-determined data generator.

Figure 14 shows the test environment for evaluation of the implemented MIPI C-PHY receiver chip. First of all, the pattern generator supplied the Flip[6:0], Rotate[6:0], and Polarity[6:0] to the MIPI C-PHY transmitter board so that the MIPI C-PHY transmitter chip generated MIPI C-PHY signals with three voltage levels. Actually, the MIPI C-PHY transmitter and PLL implemented in the MIPI C-PHY receiver chip were operated as the MIPI C-PHY transmitter chip on the MIPI C-PHY transmitter board. The MIPI C-PHY transmitter chip supplied the data for normal operation and the pre-determined data for the optimization of the adaptive level-dependent equalization to the proposed MIPI receiver test board. The function of the implemented MIPI C-PHY receiver chip was verified by measuring the synchronization word generated from the reception of the seed word and by comparing the parallel input data of the MIPI C-PHY transmitter board and the parallel output data of the MIPI C-PHY receiver board. According to the MIPI C-PHY specification V1.1, the synchronous word is generated when the Flip[6:0], Rotate[6:0], and Polarity[6:0] are 7b’0111110, 7b’1xxxxx1, and 7b’1xxxxx1, respectively, where “x” means “do not care”. Furthermore, the PO_LANE_#[16:0] is determined as [1, 1, 0, 0, 1, 0, ro6, po6, ro5, po5, ro4, po4, ro3, po3, ro0, po0] by the mapper and de-mapper of the MIPI C-PHY when the PO_LANE_#[16:0] is between 0xc800 to 0xcbff and the Flip[6:0] is 7b’0000110. Figure 15a shows the signals transmitted by the MIPI C-PHY transmitter chip for functional evaluation of the MIPI C-PHY receiver chip. The synchronization word in the MIPI C-PHY receiver chip was activated with all data of the Flip[6:2] set to high. The data streams of the Rotate[3] and Polarity[0], 0000010101110101 and 1111101010001010, were acquired as the signals of the PO_LANE1_[3] and PO_LANE1_[0], respectively, as shown in Figure 15b. Furthermore, the operation of the frame clock generator that generates the clock CLKF through frequency division of 3.5 from the recovered clock was also verified by measuring the frequency of the clock CLKF. When this experiment was performed at an symbol rate of 3.0 Gsymbol/s, the measured frequency of the clock CLKF was 428.571 MHz.

Figure 16a shows the pre-determined data transmitted from the MIPI C-PHY transmitter chip for the proposed adaptive level-dependent equalization. The measured waveforms had a symbol rate of 2.5 Gsymbol/s/lane and the same pattern of the pre-determined data shown in Figure 4. Figure 16b shows the differential output eye diagram of the MIPI C-PHY transmitter for evaluation of the MIPI C-PHY receiver chip. The MIPI C-PHY transmitter was operated at a symbol rate of 3 Gsymbol/s/lane when its supply was 500 mV on a four-inch FR-4 printed circuit board. The transmitted data had a peak-to-peak time jitter of 92 ps at the zero crossing point.

Figure 17 shows the eye diagrams measured by the loopback test on the output signals of the C-PHY RX w/ALDE including Clock Recovery. These eye diagrams were measured by supplying the three signals with a symbol rate of 3 Gsymbol/s/lane, shown in Figure 16b, from the MIPI C-PHY transmitter chip to the MIPI C-PHY receiver chip. When the proposed ALDE was not in operation, the peak-to-peak time jitters (∆t_pk-to-pk_) of the received data and recovered clock were measured to be approximately 141 ps and 96 ps, respectively, as shown in Figure 17a,b. The proposed high-speed receiver with ALDE improved the peak-to-peak time jitters of the received data and recovered clock to 129 ps and 62 ps, respectively, as shown in Figure 17c,d. In this experiment, the improvement of the time jitter of the recovered clock was greater than that of the received data. It was judged that the signal integrity of the received data, which is a random signal, deteriorated compared to that of the recovered clock, which is a toggle signal, by the output driver of the loopback test. Although the improvement of the time jitter for the received data was not significant, the proposed ALDE resulted in the received data and the recovered clock having a bi-modal distribution and a normal Gaussian distribution, respectively. Thus, the peak-to-peak time jitter of the recovered clock was improved by 35.4%. In addition, the duty cycle ratio of the recovered clock was improved from 42.8% to 48.3% by applying the proposed ALDE.

Figure 18 shows the performances of the recovered clock measured according to the digital control code for the ALDE, DEL_CONT[2:0]. These measurements were performed at a symbol rate of 2.5 Gsymbol/s/lane. When the DEL_CONT[2:0] was 3b’000, the recovered clock had a bimodal distribution and a duty cycle ratio ranging from 48% to 56%, as shown in Figure 18a. Due to this, it had a large peak-to-peak time jitter of 138.8ps. As the DEL_CONT[2:0] increased, the recovered clock had a normal Gaussian distribution, and its duty cycle ratio and time jitter were improved, as shown in Figure 18a–c. The proposed ALDE optimized the performance of the recovered clock by setting the DEL_CONT[2:0] to 3b’101. Table 1 shows that the proposed receiver chip fully supports the MIPI C-PHY V1.1 specification with performances comparable to previous studies.

## 5. Conclusions

The 3.0 Gsymbol/s/lane receiver with ALDE was proposed and implemented to support the MIPI C-PHY version 1.1 specification for CMOS image sensor interface applications. The proposed ALDE improved the signal integrity of the received data and recovered clock by adjusting the duty cycle ratio of the clock recovered from the received data to 50%. Additionally, the pre-determined data transmitted from the MIPI C-PHY transmitter was proposed to perform adaptive level-dependent equalization. The area and power consumption of one lane for the MIPI C-PHY receiver were 0.097 mm^2^ and 4.9 mW/Gsymbol/s/lane, respectively. By applying the proposed high-speed receiver with ALDE, the time jitters of the received data and recovered clock had a bi-modal distribution and a normal Gaussian distribution, respectively. Thus, the peak-to-peak time jitter and duty cycle ratio of the recovered clock were improved by 35.4% and 5.5%, respectively.

## Figures and Tables

**Figure 1 sensors-21-05197-f001:**
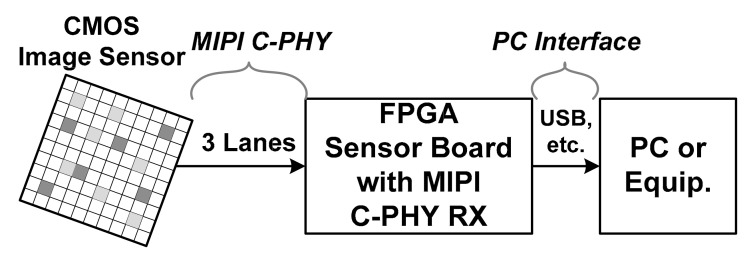
High-speed interface of FPGA-based frame grabber for CMOS image sensor.

**Figure 2 sensors-21-05197-f002:**
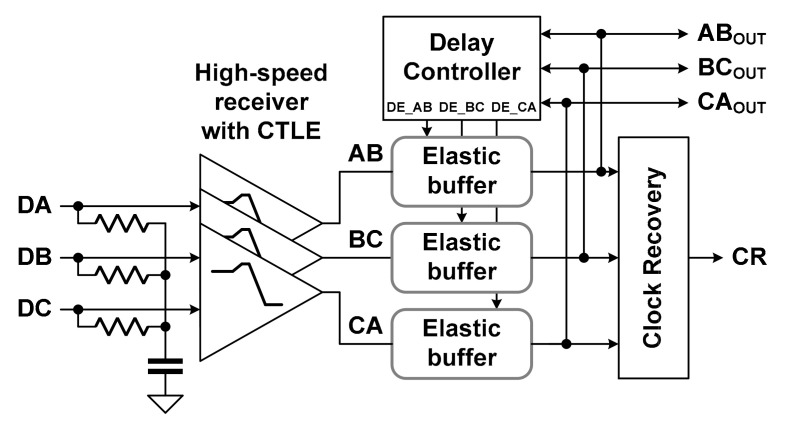
Conventional high-speed receiver using level-dependent elastic buffer for MIPI C-PHY.

**Figure 3 sensors-21-05197-f003:**
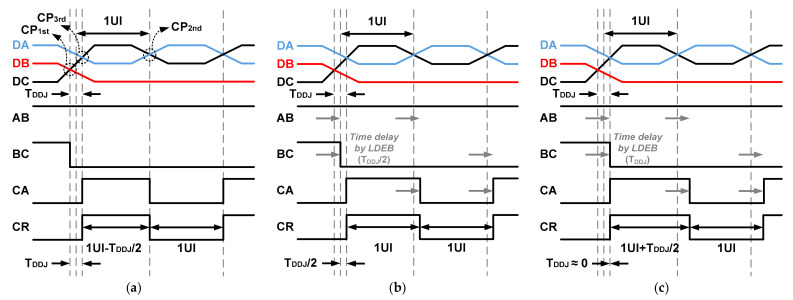
Data-dependent time jitter of data and clock in MIPI C-PHY: (**a**) without LDEB; (**b**) with optimal LDEB; (**c**) with excessive LDEB.

**Figure 4 sensors-21-05197-f004:**
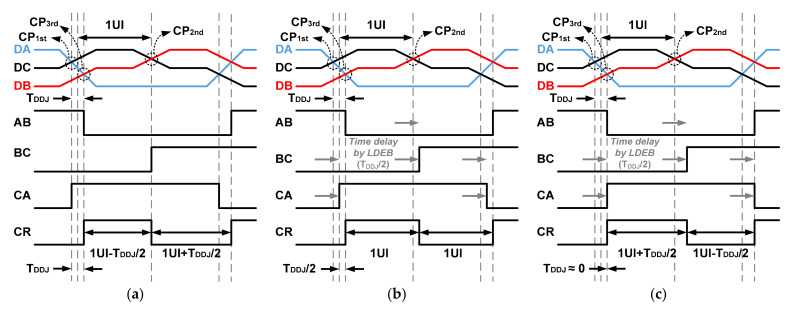
Time jitter of data and clock according to proposed pre-determined data pattern: (**a**) without LDEB; (**b**) with optimal LDEB; (**c**) with excessive LDEB.

**Figure 5 sensors-21-05197-f005:**
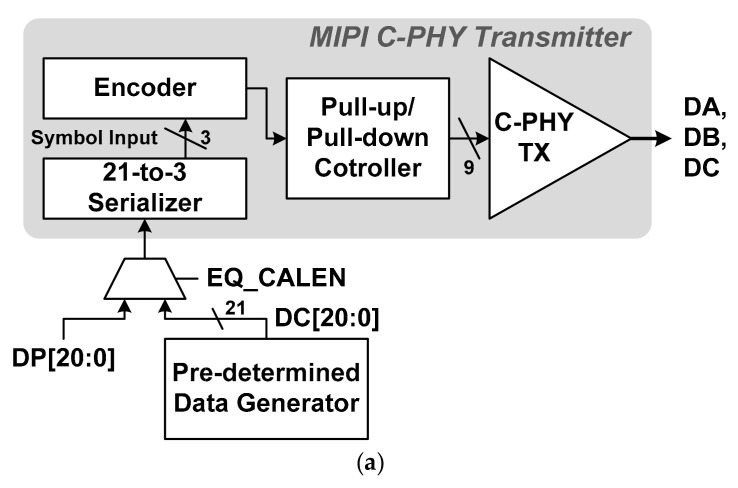
Pre-determined data generator: (**a**) Connection between MIPI C-PHY transmitter and pre-determined data generator; (**b**) Block diagram for pre-determined data generator; (**c**) Time diagram of pre-determined data generator.

**Figure 6 sensors-21-05197-f006:**
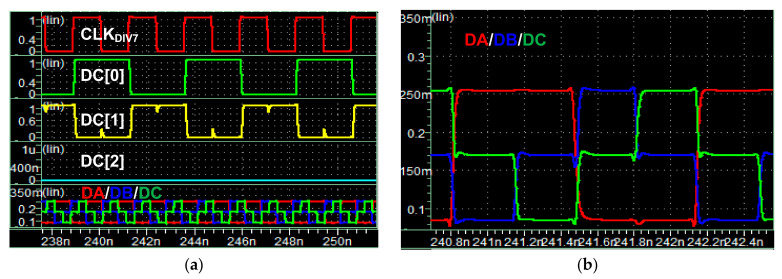
Simulation results of pre-determined data generator and MIPI C-PHY transmitter: (**a**) Waveforms of DC signals and DA/DB/DC signals; (**b**) Magnified waveforms of DA/DB/DC signals.

**Figure 7 sensors-21-05197-f007:**
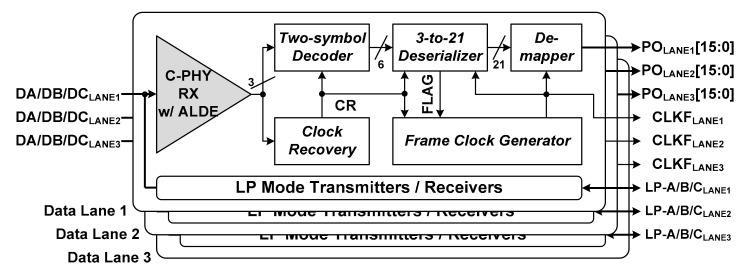
Block diagram of proposed MIPI C-PHY receiver chip.

**Figure 8 sensors-21-05197-f008:**
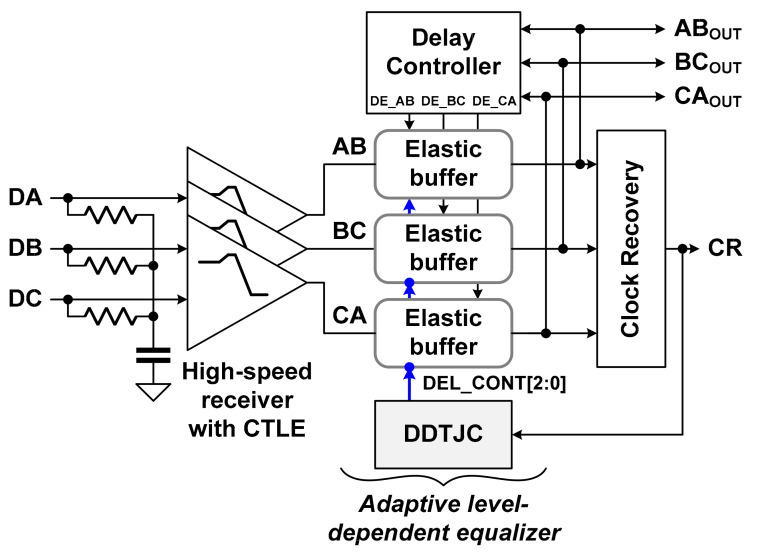
Block diagram of MIPI C-PHY receiver with ALDE.

**Figure 9 sensors-21-05197-f009:**
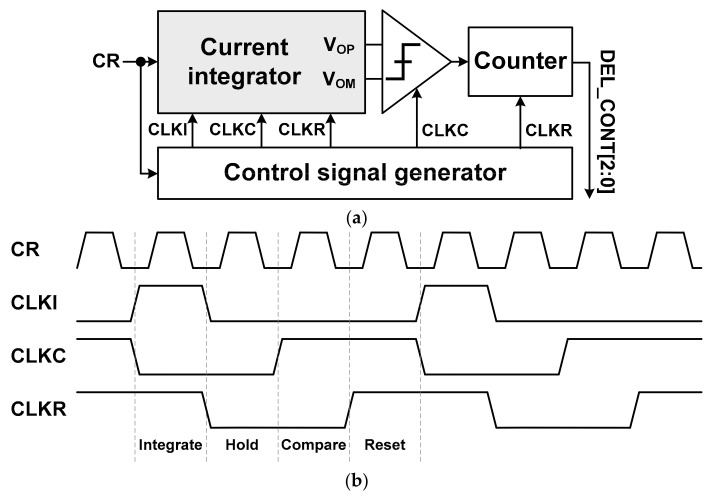
Data-dependent time jitter corrector: (**a**) Block diagram; (**b**) Time diagram.

**Figure 10 sensors-21-05197-f010:**
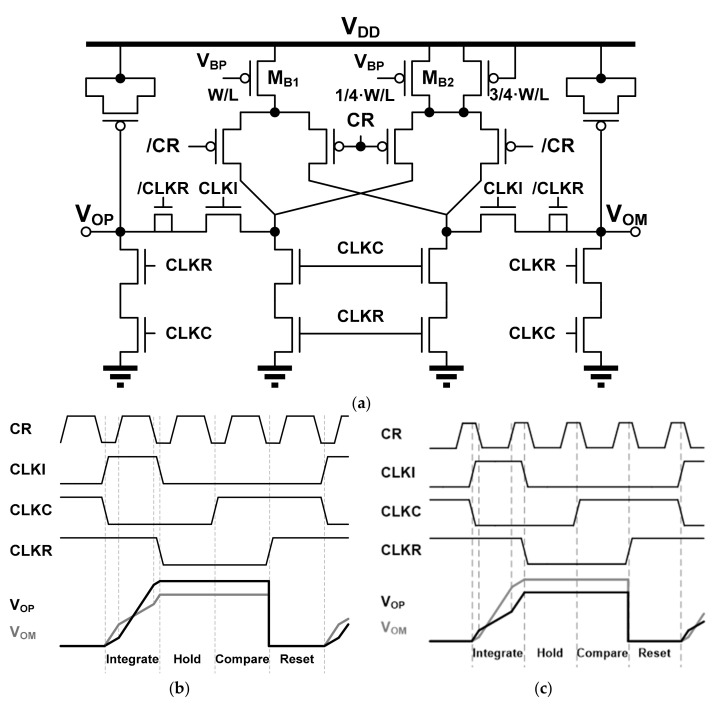
Current integrator: (**a**) Circuit diagram; (**b**) Output waveforms for CR with duty cycle ratio greater than 50%; (**c**) Output waveforms for CR with duty cycle ratio less than 50%.

**Figure 11 sensors-21-05197-f011:**
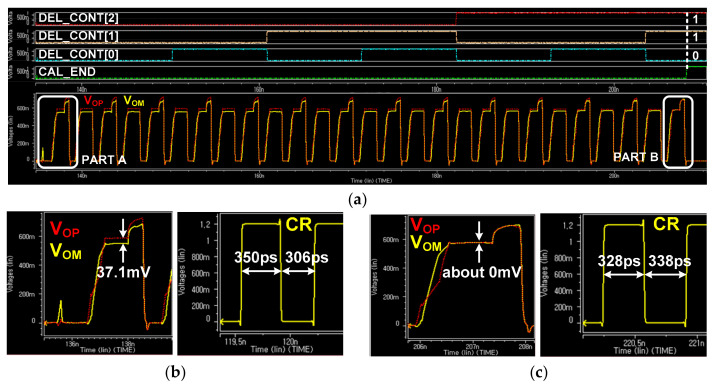
Simulation results of ALDE for pre-determined data: (**a**) Output waveforms of current integrator and output codes of DDTJC; (**b**) Waveforms of V_OP_, V_OM_, and CR in PART A; (**c**) Waveforms of V_OP_, V_OM_, and CR in PART B.

**Figure 12 sensors-21-05197-f012:**
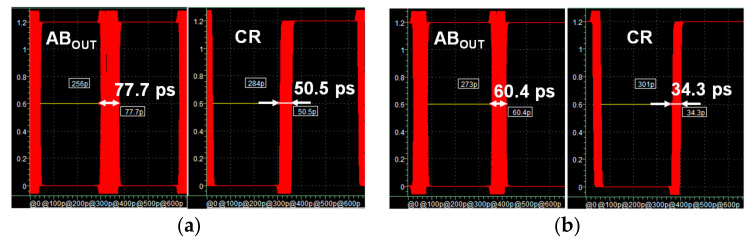
Simulation results of ALDE for random data: (**a**) AB_OUT_ and CR without adaptive level-dependent equalization; (**b**) AB_OUT_ and CR with adaptive level-dependent equalization.

**Figure 13 sensors-21-05197-f013:**
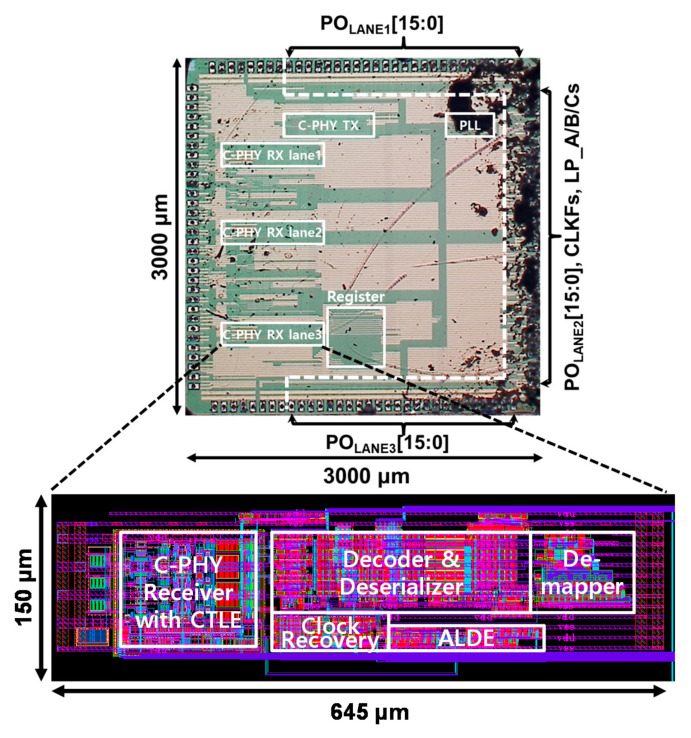
Microphotograph of implemented MIPI C-PHY receiver chip.

**Figure 14 sensors-21-05197-f014:**
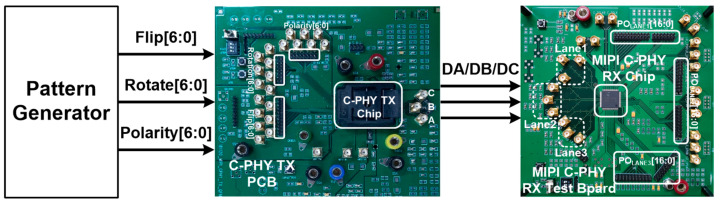
Test environment for evaluation of MIPI C-PHY receiver chip.

**Figure 15 sensors-21-05197-f015:**
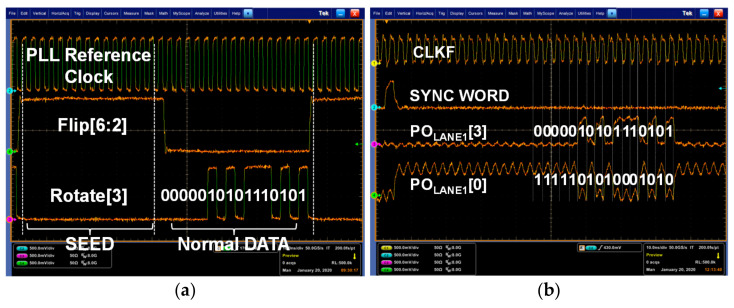
Measurement results for function evaluation of MIPI C-PHY receiver chip: (**a**) Data supplied from MIPI C-PHY transmitter chip; (**b**) Output data of MIPI C-PHY receiver chip.

**Figure 16 sensors-21-05197-f016:**
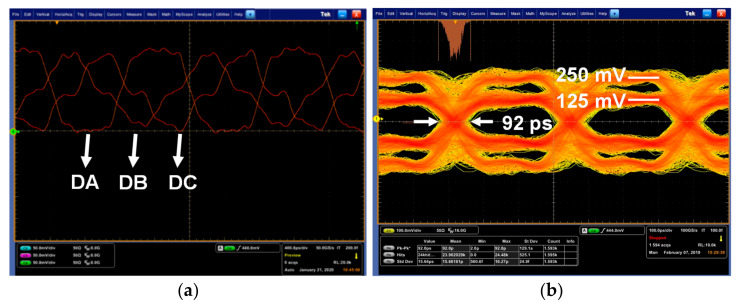
Measured waveforms of MIPI C-PHY signals transmitted from transmitter: (**a**) Pre-determined data for ALDE; (**b**) Eye diagram of MIPI C-PHY transmitter.

**Figure 17 sensors-21-05197-f017:**
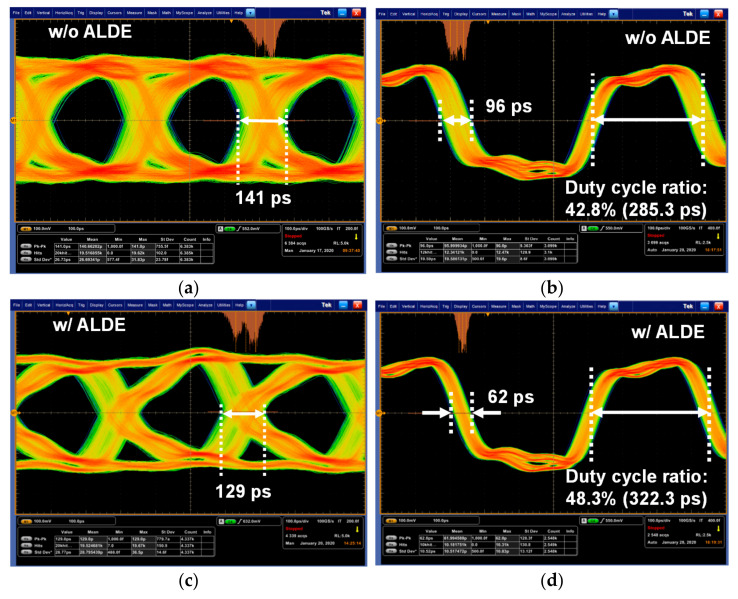
Measured eye diagrams of received data and recovered clock: (**a**) Received data without ALDE; (**b**) Recovered clock without ALDE; (**c**) Received data with ALDE; (**d**) Recovered clock without ALDE.

**Figure 18 sensors-21-05197-f018:**
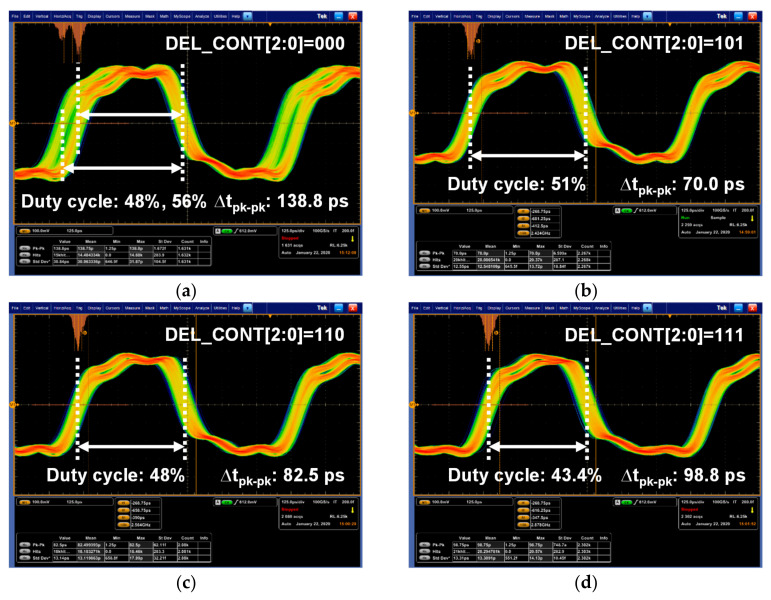
Recovered clock measured according to DEC_CODE[2:0]: (**a**) DEC_CODE[2:0] = 3b’000; (**b**) DEC_CODE[2:0] = 3b’101; (**c**) DEC_CODE[2:0] = 3b’110; (**d**) DEC_CODE[2:0] = 3b’111.

**Table 1 sensors-21-05197-t001:** Performance summary and comparison of proposed MIPI C-PHY receiver.

Reference		ISSCC’17 [7]	JSID’20 [17]	This Work
Application		Memory interface	MIPI D-/C-PHY	MIPI C-PHY V1.1
Technology		65 nm CMOS	28 nm CMOS	65 nm CMOS
Supply		1.0 V	1.0 V	1.2 V
Operation mode		Receiver	Receiver	Receiver
Data rate per lane		6.85 Gsymbol/s (15.62 Gb/s)	2.2 Gsymbol/s (5.02 Gb/s)	3.0 Gsymbol/s (6.84 Gb/s)
Function blocks for HS mode		Receiver	Receiver Clock recovery Symbol decoder Deserializer	Receiver Clock recovery Symbol decoder Deserializer De-mapper Parallel transmitter
LP mode		No	Yes	Yes
Channel for measurement		4 cm FR4	15.24 cm PCB	10.16 cm FR4
Eye opening of data	w/o Eq.	closed	-	0.58 UI
	w/ Eq.	0.51 UI	0	0.61 UI
∆t_pk-to-pk_ of recovered clock	w/o Eq.	N/A	-	96 ps
	w/ Eq.	N/A	-	62 ps
Power per lane [mW/Gb/s]		- 0.22 ^1^	1.96 ≈1.06 ^1^	2.15 0.54 ^1^

^1^ values for receiver circuit only.

## Data Availability

Not applicable.

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
