# Peer review of "A 3.0 Gsymbol/s/lane MIPI C-PHY Receiver with Adaptive Level-Dependent Equalizer for Mobile CMOS Image Sensor"

_sensors, 2021, doi:10.3390/s21155197_

Round 1

Reviewer 1 Report

The authors propose MIPI C-PHY receiver with adaptive level-dependent equalizer for mobile CMOS image sensor. The paper is well written and easy to follow, quite detailed and well structured. The results presented show the advantage of the proposed design with respect to level-dependent elastic buffer (LDEB) implementation. My comments for the authors are: 

  • Introduction could be enhanced in order to provide several approaches for the reader.
  • Implementation results could be extended in order to provide comparison results with the LDEB methodologies.

Author Response

First of all, authors would like to thank the Assistant Editor and Reviewers, for providing good comments to improve the quality of the manuscript. The authors tried their best to reply to the comments received through the paper review and to prepare the revised manuscript. In addition, the authors improved the English quality of the revised manuscript by reviewing and revising the English editing service provided by the MDPI.

Reviewer 2 Report

The authors reported a MIPI C-PHY receiver for a mobile CMOS image sensor interface. Adaptive level-dependent equalization is proposed in this paper as opposed to level-dependent elastic buffer by the same group in the prior publication. The paper is well written and verified with measurement results, which show improved time jitter and duty cycle ratio. 

I have some comments as follow:

1) The URLs of reference [5] and [6] seem to be mixed up.

2) I would suggest including the data rate in Gbps so that the readers can easily compare it with the data rate of a CMOS image sensor. 

Author Response

(The authors gave the same response as above.)

Reviewer 3 Report

  1. C-PHY targets extremely low-power applications. For instance, the attached references achieve 0.5pJ/bit and 1.9pJ/bit respectively. However, the C-PHY receiver presented in this work shows 5pJ/bit efficiency, which is even worse than various normal SerDes designs. For Therefore, I do not find a good merit of this work. 
    [r1] W. Choi, T. Kim, J. Shim, H. Kim, G. Han and Y. Chae, "23.8 A 1V 7.8mW 15.6Gb/s C-PHY transceiver using tri-level signaling for post-LPDDR4," 2017 IEEE International Solid-State Circuits Conference (ISSCC), 2017, pp. 402-403
    [r2] Kim, T-J, Hwang, J-I, Lee, S, et al. A 14-Gb/s dual-mode receiver with MIPI D-PHY and C-PHY interfaces for mobile display drivers. J Soc Inf Display. 2020; 28: 535– 547
  2. The term "adaptive" is very confusing because the authors propose to use the pre-determined data. In my opinion, it is nothing but dedicating a training period, which can never be an adaptive calibration. 
  3. It is not clear how the skew between DA/DB/DC is calibrated. If the skew is not fully calibrated, the proposed LDEB technique would not work. 
  4. Clock recovery circuits should be described. 
  5. I do not see a comparison table with other works. 

Author Response

(The authors gave the same response as above.)

Round 2

Reviewer 3 Report

I have no further comments.